# Nutritional Quality of Plant-Based Meat Products Available in the UK: A Cross-Sectional Survey

**DOI:** 10.3390/nu13124225

**Published:** 2021-11-25

**Authors:** Roberta Alessandrini, Mhairi K. Brown, Sonia Pombo-Rodrigues, Sheena Bhageerutty, Feng J. He, Graham A. MacGregor

**Affiliations:** Wolfson Institute of Population Health, Barts and the London School of Medicine and Dentistry, Queen Mary University of London, London EC1M 6BQ, UK; mhairi.brown@qmul.ac.uk (M.K.B.); s.pombo@qmul.ac.uk (S.P.-R.); s.bhageerutty@qmul.ac.uk (S.B.); f.he@qmul.ac.uk (F.J.H.); g.macgregor@qmul.ac.uk (G.A.M.)

**Keywords:** plant-based meat, alternative protein, vegetarian, nutrient profile, salt

## Abstract

Plant-based meat (PBM) has been gaining popularity due to increasing concerns over health, animal welfare, and environmental issues linked to animal foods. This study aimed to compare the nutrient profile of PBM with equivalent meat products. We conducted a cross-sectional survey of 207 PBM and 226 meat products available from 14 retailers in the UK. We extracted data on energy density, total and saturated fat, protein, fiber, and salt per 100 g from product packaging and calculated the nutrient profile of each product. Compared to meat, PBM had significantly lower energy density, total fat, saturated fat, protein, and significantly higher fiber. Salt content was significantly higher in five out of six PBM categories. Based on the UK’s Nutrient Profiling Model, 14% of PBM and 40% of meat products were classified as “less healthy” (*p* < 0.001). When considering the UK’s front-of-pack labelling criteria 20% of the PBM and 46% of meat products were considered high in either total fat, saturated fat, or salt (*p* < 0.001). Nearly three quarters of PBM products did not meet the current UK salt targets. PBM products have a better nutrient profile compared to meat equivalents. However, more progress is needed to reduce salt in these products.

## 1. Introduction

Eating more plant-based foods in place of meat has been widely recommended for health and environmental benefits [1]. Between 2017 and 2019, the proportion of the UK population attempting to reduce their meat consumption rose from 28% to 39% [2]. Those looking for alternative protein sources are increasingly turning to plant-based meat (PBM), which are products resembling the taste, appearance, and texture of widely consumed meat products such as sausages and burgers. In 2018, the UK launched more PBM products than any other country, with global sales predicted to reach 10% of the total meat market by 2029 [3,4]. The rapid increase in consumption of PBM has raised questions on their overall healthiness. Some authors have classified PBM products as ultra-processed food products [5], which are defined as “formulations of substances derived from foods, such as starches, sugars, fats, and protein isolates, with little, if any, whole food, and often with added flavors, colors, emulsifiers, and other cosmetic additives” [6]. Foods belonging to the ultra-processed food category include popular products such as biscuits, sausages, and chocolate confectionery. Evidence shows a strong association between ultra-processed food consumption and increased risk of obesity and non-communicable diseases (NCDs) [7,8]. However, there is no evidence of a relationship between PBM consumption and increased risk of obesity and NCDs.

Another way to consider the overall healthiness of manufactured food products such as PBM is to look at their calorie and nutrient content and overall nutrient profile. Nutrient profiling tools, such as the UK’s Nutrient Profiling Model, assess the overall healthiness of food products through the calculation of a nutrient profiling score (NPS). The NPS considers the balance of fiber, fruit and vegetable, and protein content against energy, sugars, saturated fat, and salt (sodium) content [9]. Based on their NPS, products can then be defined as “healthier” or “less healthy” [9]. The NPS is currently used to identify products that cannot be advertised during children’s television programs in the UK (i.e., if products have a NPS above a specified threshold, they are deemed ‘less healthy’ and are not permitted to be advertised to children) [9]. The UK’s front of pack (FoP) nutrition labeling scheme can also give an indication of the healthiness of a product, displaying levels of total fat, saturated fat, sugars, and salt according to thresholds for each nutrient [10]. To date, only a few studies have comprehensively assessed the healthiness of PBM products by considering their calorie and nutrient content [11,12,13,14], and no study has compared the overall nutrient profile of a representative sample of PBM and the corresponding meat equivalents currently available in the UK.

A common issue regarding the nutritional quality of PBM (as for all manufactured food products) is their high salt content. In 2018, Action on Salt carried out a survey of meat alternatives that raised concerns over these products’ high salt levels [15]. Excess salt intake increases the risk of high blood pressure, cardiovascular disease (CVD), osteoporosis, kidney disease, and stomach cancer [16]. The UK’s current population salt intake (8.4 g/day) is 40% higher than the recommended maximum of 6g/day, and around three-quarters of salt consumed by the population comes from manufactured food [17]. To tackle excessive salt intake in the population, the UK Government set voluntary salt targets for more than 70 manufactured food categories, including meat alternatives, for the food industry to achieve in a given timeframe [18]. The targets have been in place since 2006 and are regularly reviewed and reset to progressively lower levels to achieve a gradual and stepwise reduction in salt content. The most recent targets are due to be met by the end of 2024 [18]. 

The primary objective of this study is to compare the nutrient content of PBM with the nutrient content of their corresponding meat products. The nutrients examined in this evaluation focus on those important for prevention of chronic disease such as obesity, type 2 diabetes, and CVD. The second objective of this study is to compare the nutritional profile of PBM products available in UK retailers using the UK’s Nutrient Profiling Model and FoP labeling criteria and compare it with the nutritional profile of corresponding meat products. The third objective of this study is to compare the salt content of PBM products with the current UK salt reduction targets.

## 2. Materials and Methods

### 2.1. Plant-Based Meat Data 

We carried out a cross-sectional survey of PBM products in January 2020. We collected data from Aldi, Asda, Iceland, Lidl, Morrisons, Sainsbury’s, Tesco, The Co-operative, and Waitrose. These nine supermarket chains jointly hold more than 90% of the grocery market share in the UK [19]. We also collected data from Marks & Spencer, Boots, Holland & Barrett, Whole Foods, and Planet Organic as these chains are known for offering a wide range of PBM products. 

We extracted nutrient data per 100 g from nutrition information panels on products. Under the EU labeling legislation, nutritional information for products where a nutrient is negligible is allowed to be labeled as “trace” or provided with “<”. In these circumstances, where any nutrient was displayed as “trace”, this was replaced with 0. Similarly, where the nutrient content was <0.01, this was replaced with 0.01, <0.1 was replaced with 0.1, and so on. We collected the data on one occasion for each supermarket in the London metropolitan area. To obtain the largest possible sample, large stores were chosen instead of smaller stores. 

#### 2.1.1. Inclusion and Exclusion Criteria for Plant-Based Meat products 

We selected products according to the following inclusion criteria:Products designed to mimic the taste, texture, and full consumer experience of meat that are made of fungal or plant-based ingredients;Chilled, frozen, and ambient PBM products;Vegetarian PBM products (i.e., those with the presence of eggs and or dairy in the ingredients list);PBM products whose name or description included nouns generally used for meat products (e.g., burger, sausage);We considered only one example of one formulation regardless of the different pack sizes.

We excluded products based on the following criteria:Products that have not been designed to mimic meat, such as tofu, tempeh, falafels, and vegetable fritters;Vegan cheese and ready meals based on PBM as these categories require separate assessment.

#### 2.1.2. Product Categorization

The plant-based meat products were categorized according to their description into six main groups (sausages, burgers, plain poultry alternatives, breaded poultry alternatives, mince, and meatballs) to be compared with their respective meat categories (Table 1). Thirty-nine products did not fit in any of the set categories and as such, were not included in the nutrient comparison but were included in the salt target analysis. We also collected data for plant-based bacon, but due to small sample size, we excluded these products from the nutrient comparison and only included them in the salt target analysis. Figure 1 shows the selection process used for the inclusion of PBM and meat products in the nutrient profiling analyses and comparisons with the most recent UK salt targets. 

### 2.2. Meat Data 

At the time of data collection (May 2020), a comprehensive review of meat products available in the UK retail market was not possible due to national lockdown restrictions following the COVID-19 pandemic. Instead, we collected nutrition information for the meat categories online from Tesco, the retailer with the largest market share in the UK [19]. We chose the meat categories to correspond with six PBM categories (sausages, burgers, plain poultry, breaded poultry, mince, and meatballs). We included processed and unprocessed products made of lamb, beef, pork, chicken, and turkey meat (e.g., chicken breast, turkey drumsticks). 

### 2.3. Statistical Analysis 

#### Nutrient Content

We reported descriptive statistics (mean, SD, range) for total fat, saturated fat, fiber, protein, salt (g/100 g) and energy density (kcal/100 g) for all PBM and their corresponding meat categories. To assess differences in the mean nutrient content of PBM and the corresponding meat products, we carried out two-samples *t*-tests. 

### 2.4. Nutrient Profile

To assess the products’ nutrient profile, we used the UK’s Nutrient Profiling Model [9]. The model uses a simple scoring system where points are allocated based on the nutrient content per 100 g of food. The final product score is obtained by subtracting points for “positive” nutrients and ingredients (i.e., fiber, protein, vegetables, fruit, nut, and legume) from “negative” nutrients (i.e., energy, saturated fat, total sugar, and salt). If a food scores four or more points, it would be deemed high in fat, salt and/or sugar and therefore “less healthy”; if a product scores less than four points, it would be classified as “healthier”. Seven products did not report the amount of fiber per 100 g (which is needed to calculate the NPS). Therefore, for those products we calculated the mean fiber content in each PBM category and applied it to the calculation to obtain a complete score. To calculate the points for fruit, vegetable, nuts, legumes (FVNL) we manually extracted data on the percentage of these ingredients from the ingredient list of each product assigning one “positive” point if FVNL was above 40% of the ingredient composition or if four out of five first ingredients listed were FVNL. We did not include texturized plant-based protein such as pea or soya in FVNL points calculations as these ingredients are usually used to replace animal protein and not legumes and vegetables. We reported the percentage of products that would be deemed as “healthier” (i.e., NPS < 4) or “less healthy” (i.e., NPS ≥ 4). To determine whether the difference between the two percentages was statistically significant, we performed a *t*-test for proportions. 

### 2.5. FoP Labeling

We used guidance from the UK’s FoP labeling to assess the content of total fat (≤3 g/100 g as low, >3 g to ≤17.5 g/100 g as medium, >17.5 g/100 g as high), saturated fat (≤1.5 g/100 g as low, >1.5 g to ≤5 g/100 g as medium, >5 g/100 g as high) and salt (≤0.3 g/100 g as low, >0.3 to ≤1.5 g/100 g as medium, >1.5 g/100 g as high) in each product. We did not check for sugar criteria as the sugar content in both PBM and meat products was low. We calculated the percentage of PBM and animal meat products that would be considered high in at least one nutrient (either total fat, saturated fat, salt, or a combination of these). To determine whether the difference between the two percentages was statistically significant, we performed a *t*-test for proportions. All the analyses were performed using R version 4.0.0 (R Foundation for Statistical Computing).

### 2.6. Comparison with UK’s 2024 Salt Reduction Targets

We compared the salt content of PBM against the most recent salt reduction targets, which included specific targets for the following PBM categories:Plain meat alternatives: 0.63 g salt per 100 g (maximum target);Meat-free products: 1.19 g/100 g (maximum target);Meat-free bacon: 1.78 g/100 g (maximum target);

We considered only the products reporting nutrition information “as sold” as this is the format of the UK salt targets. Products were categorized according to the classification used in the targets, as “plain meat alternatives” including plain mince or plain chicken pieces. The “meat-free products” and “meat-free bacon” included most of the remaining meat alternative products.

## 3. Results

Nutrient data were collected from 207 PBM products. Of these, 168 products were grouped into six categories (sausages, burgers, plain poultry alternatives, breaded poultry alternatives, mince, and meatballs). Nutritional data were collected from 226 comparable meat products. Figure 1 shows the product’s selection process into the different analyses performed.

### 3.1. Nutrient Content per 100 g (n = 168; n = 226)

*Energy density:* when compared to their equivalent meat products, energy density was significantly lower in four out of six PBM categories (Table 2);*Total fat:* total fat content was significantly lower in four out of six PBM categories;*Saturated fat*: saturated fat was significantly lower in five out of six PBM categories.Overall, meat products had more than twice the amount of saturated fat compared to PBM products. For example, the average saturated fat content in meat-based burgers was almost four times higher than in plant-based burgers (6.6 ± 2.7 g vs. 1.7 ± 1.5 g, *p* < 0.001) (Table 2);*Fiber:* the fiber content of all PBM categories was significantly higher (*p* < 0.001) than the fiber content of the corresponding meat-based products. Overall, PBM categories had around four times more fiber than their corresponding meat products (Table 2);*Protein:* the protein content in PBM products was significantly lower than meat-based products in four out of six categories. Plant-based and meat-based breaded poultry and mince had similar protein content. Burgers had the largest difference in protein content; 11.1± 6.1 in meat vs. 19.9 ± 3.8 in plant-based burgers (*p* < 0.001) (Table 2);*Salt:* the salt content was significantly higher in five out of six PBM categories. Plant-based sausages had a similar salt content to their meat counterparts. The plain plant-based poultry category had more than twice the amount of salt than plain poultry (1.3 ± 0.6 vs. 0.5 ± 0.3, *p* < 0.001) (Table 2);There was a wide variation in the energy density and nutrient content per 100 g of all PBM and meat categories.

### 3.2. Overall Nutrient Profile (n = 168; n = 226)

#### 3.2.1. UK’s Nutrient Profiling Model

The percentage of products that had an NPS ≥ 4 and thus classified as “less healthy” was 13.7% (*n* = 23) for PBM compared to 40% (*n* = 91) for meat products (*p* < 0.001). Compared to meat products, the average NPS of PBM was significantly lower than their meat counterparts, except for plant-based plain poultry (Table 3). The largest difference observed between PBM and meat-based products was for sausages, with an average difference of 12 NPS points (1.7 ± 5.9 vs. 13.8 ± 5.5, *p* < 0.001) (Table 3). 

#### 3.2.2. Front-of Pack Labelling

Overall, 20% (*n* = 34) of PBM products and 46% (*n* = 104) of meat products had at least one high criteria for total fat, saturated fat, and/or salt (*p* < 0.001). The largest difference was seen in saturated fat content, where four percent (*n* = 7) of PBM products were deemed high in saturated fat, compared to 41.7% (*n* = 94) of meat products (*p* < 0.001). Similarly, around five percent (*n* = 9) of PBM products and 30% (*n* = 68) of meat products were high in total fat (*p* < 0.001). Conversely, around 15% (*n* = 26) of PBM products and 13.7% (*n* = 30) of meat products would be high in salt (*p* = 0.63). 

#### 3.2.3. Comparison with the 2024 Maximum Salt Targets (*n* = 123) 

Of the 207 PBM products with nutritional data, 123 reported nutrition information as sold and could be compared with the maximum salt targets (Figure 1). These products were categorized into the three meat alternatives categories: plain meat alternatives, meat-free products, and meat-free bacon [18]. Overall, around a quarter of the surveyed products had a salt content per 100 g below their respective maximum salt reduction targets (Table 4). 

## 4. Discussion

Very few published studies assessed the nutritional profile and overall healthiness of PBM [11,12,13,14]. The results of this first cross-sectional survey comparing the nutrient profile of PBM products available in the UK market to their equivalent meat categories support the common perception that PBM is a healthier alternative to animal products from a chronic disease prevention perspective [20]. The majority of PBM products scored more favorably than their meat counterparts, with fewer products being classified as “less healthy” (13.7% vs. 40%, respectively) as defined by the UK’s Nutrient Profiling Model. Similarly, classification via the UK’s FoP labeling demonstrated that fewer PBM products would be considered high in total fat, saturated fat and/or salt compared to meat products (20% vs. 46%, respectively). 

Consuming PBM instead of meat might improve the overall environmental sustainability of diets [21], positively impact animal welfare as it might reduce dependence on livestock [22], and have positive public health implications. Our analysis showed that most PBM categories had a lower energy density than their corresponding meat products. The lower energy density of PBM can be attributable to their lower fat and higher dietary fiber content. Substituting meat products with PBM could therefore be potentially used as a weight management tool to reduce excess consumption of calories and reduce incidences of overweight and obesity. This is supported by some evidence from a randomized crossover trial which showed a significant weight loss in the phase in which participants consumed PBM instead of meat [23]. 

Our data showed that saturated fat was also significantly lower in PBM. In the UK and many other European countries, saturated fat intake exceeds the recommended levels of 10% of the total energy [24,25]. Both the UK Scientific Advisory Committee on Nutrition and the World Health Organization have recently issued recommendations urging countries to implement strategies to reduce saturated fat intake at the population level due to its effects on low-density lipoprotein cholesterol [26,27]. Consuming PBM instead of animal meat could potentially be used as a strategy to reduce saturated fat intake and thereby cardiovascular disease risk. Dietary fiber content was significantly higher in PBM. In the UK and in most European countries, fiber intake is below the recommended intake [28], and eating PBM could help consumers achieve their recommended fiber intake. 

Our results indicate that most meat categories have a less favorable nutrient profile compared to PBM. Manufacturers could improve their meat products’ nutritional quality by altering a product’s composition to include more vegetables, legumes, pulses, or texturized plant-based protein [29]. This practice is feasible and is already in operation, with a lamb mince product blended with vegetables currently available in the UK’s largest retailer. This product has 32% fewer calories and half the fat and saturated fat content than standard lamb mince from the same brand and retailer. Reformulating products in this way could also help manufacturers reduce their environmental impact while meeting the taste preferences of a wider number of consumers [29,30]. 

Whilst PBM scored more favorably on total fat, saturated fat, and dietary fiber content, most PBM products had a higher salt content than their meat counterparts. Plant-based plain poultry was exceptionally high in salt, with most products exceeding their respective salt target. Overall, nearly three quarters of PBM analyzed exceeded their respective salt targets. This finding indicates that more progress in salt reduction is needed in the PBM industry. Globally and in the UK, salt intake is one of the leading causes of mortality and morbidity and reducing salt in food is widely recognized as a cost-effective approach to improve public health [16,31]. Our results also showed a large variation in the salt content of PBM, thus demonstrating that reducing salt in these products is indeed possible. PBM manufacturers have a vital role in providing consumers, restaurants, and caterers with products that do not contain excessive amounts of salt so that meals provided can be both environmentally sustainable and healthy. 

### 4.1. Comparison with Other Studies 

We compared our findings with a recent study, which reported nutrition information of 137 PBM products sold in four leading supermarket chains in Australia [11]. We found a negligible difference between Australian and UK products’ nutrient and energy content; however, salt content was on average 0.2 g/100 g higher in plant-based sausages and burgers sold in the UK. This finding demonstrates that salt reduction is indeed possible and that the UK maximum salt target of 1.19 g/100 g set for plant-based sausages and burgers could be lowered even further. Similarly to our study, the researchers also measured products’ healthiness through a nutrient profiling model, the Australian Health Star Rating, and found that, on average, PBM had a more favorable profile than meat-based products except for mince. Another recent report looked at the nutritional quality of PBM available in Ireland [12]. The report showed that the salt content of the most popular plant-based categories (burgers, sausages, and breaded chicken) was similar to the products included in the present study [12]. A further study carried out in the USA looked at the nutritional profile of some popular PBM products; however, the representativeness of its sample was questionable (i.e., the US study included only a small selection of products), and therefore, we could not make a comparison between the US and UK products [32].

### 4.2. Limitations

To evaluate overall product healthfulness, we used the NPM and the UK FoP labeling guidance, which focus on crucial nutrients for NCD prevention and do not take into account micronutrient content. Meat is a source of important micronutrients such as zinc, iron, and vitamin B12. A recent investigation carried out in the US reported that plant-based minced beef has lower levels of zinc and vitamin B12 [14]. Some nutrient profiling models also consider micronutrient content [33], but we could not use these as the current UK legislation does not require reporting micronutrient content on food packaging. 

Another important limitation of our study is that the data for meat products were only collected online from one retailer, Tesco. Tesco is the UK supermarket with the largest market share, offering a wide variety of own-label and branded products. It is plausible to consider that the meat data collected can represent the products available on the UK market. 

### 4.3. Implications

The Intergovernmental Panel on Climate Change (IPCC) stated that reducing meat consumption is essential to limit global warming to 1.5 °C above the pre-industrial levels [34]. Consuming alternative proteins such as PBM instead of meat could help consumers to shift to more sustainable diets. The dietary guidelines from some countries and those developed by the Eat Lancet commission recommend reducing meat intake on health and environmental grounds and increasing the ratio of legumes, pulses, nuts, and other “soy foods” [1]. However, these recommendations do not clearly indicate whether PBM can be consumed as a healthy alternative to conventional meat products such as burgers and sausages [1]. Our results, however, suggest that compared to meat, PBM has a more favorable nutritional profile and could reduce the intake of excess calories and nutrients linked to obesity and CVD, thus producing positive health impacts in the long term. Nevertheless, further evidence from trials and epidemiological studies investigating PBM consumption patterns and NCDs’ biomarkers are needed to establish whether PBM can improve population health [5,22]. 

On the other hand, our results also highlighted the excessive salt content of PBM available in the UK. Manufacturers could effectively improve the nutritional quality of these products by reducing their salt content. Our results showed a considerable variation in the salt content of these products, thus demonstrating that reducing salt is entirely possible. 

## 5. Conclusions

Our results show that most PBM categories have a more favorable nutrient profile than their meat counterparts, with fewer products being classified as “less healthy” (14% vs. 40%, respectively). Similarly, classification via the UK’s FoP labeling demonstrated that fewer PBM products scored high for total fat, saturated fat and/or salt than meat products (20% vs. 46%, respectively). However, salt content in PBM was high, with only almost a quarter of products meeting the most recent salt targets. Manufacturers could effectively improve the nutritional quality of these products by reducing their salt content. PBM products are a rapidly growing food category, and more evidence from epidemiological studies and trials is needed to investigate their long-term impacts on health.

## Figures and Tables

**Figure 1 nutrients-13-04225-f001:**
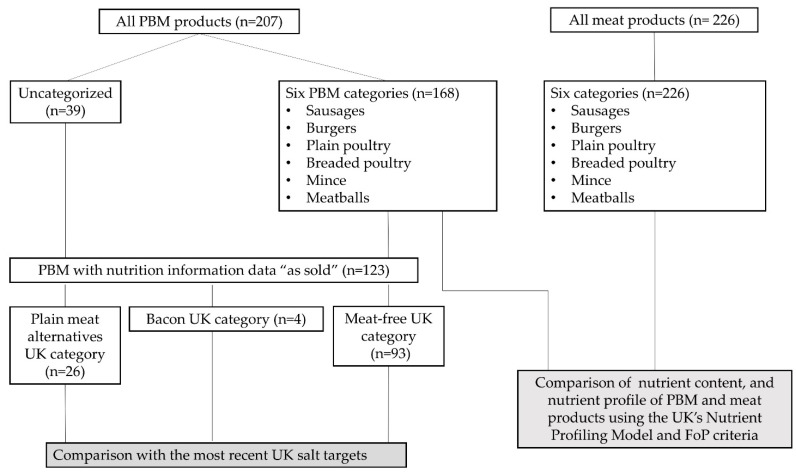
The selection process used for the inclusion of PBM and meat products in the nutrient profiling analyses and comparisons with the most recent UK salt targets.

**Table 1 nutrients-13-04225-t001:** Classification of PBM categories.

Category	Description
Sausages	Meat-free “sausages”, “brats”, “bangers”, “hot dogs”, “wieners”, “polony”, “shroomdogs”
Burgers	Meat-free “burgers”, “quarter pounders”, “patties”
Plain poultry	Meat-free chicken mimicking chicken “strips”, “pieces”, “tenders”, and “bites”
Breaded poultry	Meat-free breaded chicken mimicking chicken “nuggets”, “goujons”, “escalopes”, “southern-style”, “crispy tenders or slices”, “schnitzels”, and “kievs”
Mince	Meat-free “mince”
Meatballs	Meat-free “meatballs” and “balls”
Bacon	Meat-free “bacon” or “rashers”

**Table 2 nutrients-13-04225-t002:** Mean energy density (kcal/100 g) and nutrient content (g/100 g) ± SD and range of meat and PBM categories. PB = plant-based.

	Energy Density Mean ± SD (kcal/100 g)	*p*	Total Fat, Mean ± SD (g/100 g)	*p*	Saturated, Mean ± SD (g/100 g)	*p*	Fibre, Mean ± SD (g/100 g)	*p*	Protein, Mean ± SD (g/100 g)	*p*	Salt, Mean ± SD (g/100 g)	*p*
Sausages (*n* = 72)	259.5 ± 51.1 (129.0–368.0)	<0.001	19.1 ± 6.4 (3.9–32.1)	<0.001	6.8 ± 2.5 (1.1–11.6)	<0.001	1.1 ± 0.4 (0.2–2.0)	<0.001	15 ± 3.1 (10.2–22.8)	0.002	1.6 ± 0.5 (0.6–2.9)	0.141
PB sausages (*n* = 41)	175.4 ± 50.0 (103.0–267.0)		8.9 ± 5.3 (1.4–20.0)		2.6 ± 3.1 (0.2 –15.8)		5.4 ± 2.0 (1.7–11.0)		12.1 ± 5.5 (2.6–25.0)		1.4 ± 0.4 (0.9–2.4)	
Burgers (*n* = 26)	236.4 ± 54.9 (129.0–315.0)	0.010	15.0 ± 6.8 (3.9–26.1)	<0.001	6.6 ± 2.7 (1.1 –11.0)	<0.001	0.9 ± 0.5 (0–2.3)	<0.001	19.9 ± 3.8 (13.7–26.4)	<0.001	0.9 ± 0.3 (0.6–1.9)	<0.001
PB burgers (*n* = 58)	203.1 ± 46.4 (120.0–338.0)		10.3 ± 5.0 (1.4–27.0)		1.7 ± 1.5 (0.2–10.0)		4.6 ± 2.1 (0.3–10.2)		11.1 ± 6.1 (3.2–23.0)		1.1 ± 0.4 (0.4–2.0)	
Plain poultry (*n* = 68)	146.8 ± 39.2 (99.0–252.0)	0.299	5.02 ± 4.7 (0.9–17.7)	0.578	1.4 ± 1.4 (0.3–6.3)	0.003	0.2 ± 0.3 (0–1.3)	<0.001	23.3 ± 5.2 (0.5–35)	<0.001	0.5 ± 0.3 (0 –1.6)	<0.001
PB plain poultry (*n* = 20)	161.5 ± 57.9 (95.0–336.0)		5.7 ± 5.0 (0.5–22.0)		0.8 ± 0.6 (0.1–2.9)		5.6 ± 2.3 (0.4–12.0)		18.7 ± 4.1 (9.1–29.7)		1.3 ± 0.6 (0.4–2.3)	
Breaded poultry (*n* = 38)	252.9 ± 29.1 (186.0–304.0)	0.026	14.0 ± 3.6 (6.8–21.0)	0.009	2.6 ± 1.7 (0.7–8.1)	<0.001	1.2 ± 0.6 (0–2.4)	<0.001	13.6 ± 2.6 (7.2–20.3)	0.522	0.7 ± 0.2 (0.2–1.2)	<0.001
PB breaded poultry (*n* = 22)	233.9 ± 31.5 (188.0–321.0)		11.0 ± 4.3 (4.3–22.0)		1.3 ± 0.9 (0–4.0)		5.1 ± 2.0 (2.0 –8.9)		13.2 ± 2.5 (9.1–17.8)		1.2 ± 0.3 (0.4 –1.8)	
Mince (*n* = 14)	183.2 ± 47.1 (114.0–252.0)	0.572	11.4 ± 6.1 (1.3–20.0)	0.006	5.1 ± 3.1 (0.4–9.8)	0.002	1.6 ± 0.5 (0–2.0)	<0.001	19.6 ± 2.9 (13.4–25.4)	0.670	0.3 ± 0.2 (0.1–0.9)	0.002
PB mince (*n* = 16)	170.3 ± 74.8 (66.0–374.0)		5.7 ± 3.7 (0.2 –11.3)		1.7 ± 2.2 (0.1–7.9)		5.6 ± 3.1 (1.6–16.0)		20.8 ± 10.2 (2.5–30.6)		0.7 ± 0.4 (0–1.5)	
Meatballs (*n* = 8)	219.6 ± 34.4 (178.0–279.0)	0.019	12.1 ± 4.1 (6.1–17.9)	0.076	4.6 ± 1.5 (1.7–6.2)	<0.001	0.8 ± 0.5 (0.1–1.7)	<0.001	21.3 ± 3.4 (16.8–25.2)	<0.001	0.8 ± 0.1 (0.1–0.9)	0.012
PB meatballs (*n* = 11)	178.9 ± 31.6 (145.0–236.0)		8.7 ± 3.3 (4.3–15.0)		1.1 ± 0.4 (0.5 –1.7)		5.5 ± 0.8 (4.0–6.7)		13.9 ± 3.0 (0.7–0.9)		1.1 ± 0.3 (0.8–1.7)	

**Table 3 nutrients-13-04225-t003:** Mean nutrient profiling scores of meat and PBM categories ± SD.

	Sausages	*p*	Burgers	*p*	Plain Poultry	*p*	Breaded Poultry	*p*	Mince	*p*	Meatballs	*p*
Meat	13.8 ± 5.5	<0.001	9.3 ± 5.6	<0.001	−1.0 ± 2.8	0.710	2.2 ± 3.6	0.003	3.2 ± 5.9	0.004	5.0 ± 3.6	<0.001
PBM	1.7 ± 5.9		0.8 ± 3.7		−1.5 ± 5.5		−0.9 ± 3.6		−3.4 ± 5.4		−4.0 ± 2.0	

**Table 4 nutrients-13-04225-t004:** Comparison with the most recent UK maximum salt targets.

Category	Grams of Salt/100 g, Mean ± SD (Min–Max)	UK Maximum Salt Target (Grams of Salt/100 g) for 2024	N (%) of Products Meeting the 2024 Target
Plain meat alternatives (*n* = 26)	1.12 ± 0.46 (0.05–2.3)	0.63	4 (15.4%)
Meat-free products (*n* = 93)	1.30 ± 0.45 (0.35–2.4)	1.19	30 (35.5%)
Meat-free bacon (*n* = 4)	2.12 ± 0.94 (0.8–2.9)	1.78	1 (25%)
Total (*n* = 123)	-	-	35 (28.5%)

## Data Availability

The data was obtained from FoodSwitch via a license with The George Institute for Global Health (TGI). Non-identifiable data can be shared on request, but a comprehensive dataset will need to be acquired from TGI.

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
