# Peer review of "Nutritional Quality of Plant-Based Meat Products Available in the UK: A Cross-Sectional Survey"

_nutrients, 2021, doi:10.3390/nu13124225_

Round 1

Reviewer 1 Report

This paper addressed a timely and important topic. The study was well designed with appropriate analyses carried out. Also, the paper is very clearly written. Areas for improvement are as follows:

Line 55- add recently published study by Harnack et al in Journal of Academy of Nutrition and Dietetics to the list of references for previous studies evaluating the nutritional quality of PBM.

Lines 70-75- the nutrients examined in this evaluation focus on those important for prevention of chronic disease such as obesity, type 2 diabetes, and CVD. Nutrients of relevance for prevention of micronutrient deficiency diseases (e.g. iron deficiency anemia, cognitive impairment due to vitamin B12 deficiency, etc.) were not examined. Consequently, this point needs to be made clear in this paragraph and throughout the discussion section of the paper.

Line 117- delete duplicate wording of 'in the'.

Table 1- how were plant-based crumbles designed as replacements for ground beef classified? Perhaps they were not available in UK marketplace, which may explain why they are not listed?

Lines 121-127- it appears that only processed/packaged meat products were included? This needs to be clarified. If only processed/packaged meats were included it will be important to be clear in the discussion section that findings have limited generalizability (can't assume same nutritional differences and similarities exist for minimally processed/fresh meats).

Line 234- Add Harnack et al. paper to list of previous studies. 

Line 237- clarify that findings suggest PBM may be a healthier alternative from a chronic disease prevention perspective.

Line 243- change 'would' to 'might'. There is research that indicates some PBM (e.g. Impossible Burger, Beyond Burger) may have a bigger or similar environmental footprint in comparison to meats such as chicken and pork. 

Lines 284-300- Integrate findings from the Harnack et al study into this section (most findings from that study were consistent with findings from your study). That study included an examination of micronutrients such as iron, zinc, and vitamin B12, and differences were found indicating PBM may be inferior to meat for some of these nutrients. It might be worth pointing this out as a way of reinforcing the focus of the present paper on evaluating nutritional quality of PBM products from a chronic disease prevention perspective.

Author Response

Response to reviewer 1

Reviewer’s comment: This paper addressed a timely and important topic. The study was well designed with appropriate analyses carried out. Also, the paper is very clearly written. Areas for improvement are as follows:

Authors’ reply: We thank reviewer 1 for appreciating our work and for providing useful comments.

Reviewer’s comment: Line 55- add recently published study by Harnack et al in Journal of Academy of Nutrition and Dietetics to the list of references for previous studies evaluating the nutritional quality of PBM.

Authors’ reply: The reference has been added to the manuscript.

Reviewer’s comment: Lines 70-75- the nutrients examined in this evaluation focus on those important for prevention of chronic disease such as obesity, type 2 diabetes, and CVD. Nutrients of relevance for prevention of micronutrient deficiency diseases (e.g. iron deficiency anemia, cognitive impairment due to vitamin B12 deficiency, etc.) were not examined. Consequently, this point needs to be made clear in this paragraph and throughout the discussion section of the paper.

Authors’ reply: we made the following changes to the objectives:

“The primary objective of this study is to compare the nutrient content of PBM with the nutrient content of their corresponding meat products. The nutrients examined in this evaluation focus on those important for prevention of chronic disease such as obesity, type 2 diabetes, and CVD”.

We also added this concept in the implications’ section:

Our results, however, suggest that compared to meat, PBM has a more favorable nutritional profile and could reduce the intake of excess calories and nutrients linked to obesity and CVD, thus producing positive health impacts in the long term.

Reviewer’s comment:  Line 117- delete duplicate wording of 'in the'.

Authors’ reply: We deleted the duplicate word “analyses”

Reviewer’s comment: Table 1- how were plant-based crumbles designed as replacements for ground beef classified? Perhaps they were not available in UK marketplace, which may explain why they are not listed?(

Authors’ reply: We checked our dataset but we could not find any product described as “crumble”.  In the UK ground beef is commonly known as mince (or minced) beef and the word mince usually indicates ground meat of any type. In our analysis we grouped all minced products together (i.e. those intended to mimic beef, pork, lamb, or poultry meat) under the category “mince”.

Reviewer’s comment: Lines 121-127- it appears that only processed/packaged meat products were included? This needs to be clarified. If only processed/packaged meats were included it will be important to be clear in the discussion section that findings have limited generalizability (can't assume same nutritional differences and similarities exist for minimally processed/fresh meats).

Authors’ reply: Most of the meat categories could be classified as processed but we also included unprocessed cuts such as chicken breast, turkey drumsticks, and fresh mince meat. So, we believe to have made a fair comparison between the PBM and the meat categories. We understand that we did not specify this in the manuscript, and we thank the reviewer for pointing this out. We made the following change:

We included processed and unprocessed products made of lamb, beef, pork, chicken, and turkey meat (e.g. chicken breast, turkey drumsticks).

Reviewer’s comment: Line 234- Add Harnack et al. paper to list of previous studies. 

Authors’ reply: The reference has been added to the manuscript

Reviewer’s comment: Line 237- clarify that findings suggest PBM may be a healthier alternative from a chronic disease prevention perspective.

Authors’ reply: we amended the manuscript as following:

The results of this first cross-sectional survey comparing the nutrient profile of PBM products available in the UK market to their equivalent meat categories support the common perception that PBM is a healthier alternative to animal products from a chronic disease prevention perspective.

Reviewer’s comment: Line 243- change 'would' to 'might'. There is research that indicates some PBM (e.g. Impossible Burger, Beyond Burger) may have a bigger or similar environmental footprint in comparison to meats such as chicken and pork. 

Authors’ reply: We have changed the “would” to “might” as the reviewer suggested.

Reviewer’s comment: Lines 284-300- Integrate findings from the Harnack et al study into this section (most findings from that study were consistent with findings from your study). That study included an examination of micronutrients such as iron, zinc, and vitamin B12, and differences were found indicating PBM may be inferior to meat for some of these nutrients. It might be worth pointing this out as a way of reinforcing the focus of the present paper on evaluating nutritional quality of PBM products from a chronic disease prevention perspective.

Authors’ reply: Unfortunately, we could not directly compare the results from Harnack et all with our results due to the different ways in which results are reported in the two studies. However, we took into account the reviewer’s comment and we added in different sections of the manuscript that our analysis focuses on critical nutrients for CVD/NCD prevention. We included this concept in the limitations section of our manuscript:  

“To evaluate overall product healthfulness, we used the NPM and the UK FoP labeling, which focus on crucial nutrients for NCD prevention and do not take into account micronutrient content. Meat is a source of important micronutrients such as zinc, iron, and vitamin B12. A recent investigation carried out in the US reported that plant-based minced beef has lower levels of zinc and vit B12”

Reviewer 2 Report

The paper of Alessandrini et al. “Nutritional quality of plant-based meat products available in the UK: a cross-sectional survey” describes an analysis of the nutrient profiles of plant-based meat products in the UK. In view of the rising market share of these products, it is a timely investigation with high relevance. The paper is based on a solid methodology, it is comprehensively written, well discussed and strength and limitations are clearly indicated. I have only minor comments to make.

Specific comments:

Line 49: Does the Nutrient Profiling Score identify products as “healthier” or “less healthy” or is only the category “less healthy” defined by the profiling model? Specify the wording.

Line 118: Check font size

Line 263: Both.. and consumers – not clear. Separate the two aspects, the manufacturer can alter the product composition, the consumer can only alter the meal composition during cooking

Line 273 – 283: plain poultry will be salted during cooking/ at the table in contrast to plant-based plain poultry. This could explain the differences in salt content. However, would a consumer individually salt plain poultry less than the processed meat alternative?

Line 302 – limitations: it is very important that you mentioned the potential differences in micronutrient content. However, you should also add the aspect of protein nutritional quality e.g. protein digestibility-corrected amino acid score and digestible indispensable amino acid score

Line 341: last punctuation mark is missing

Author Response

Response to reviewer 2

Reviewer’s comment: The paper of Alessandrini et al. “Nutritional quality of plant-based meat products available in the UK: a cross-sectional survey” describes an analysis of the nutrient profiles of plant-based meat products in the UK. In view of the rising market share of these products, it is a timely investigation with high relevance. The paper is based on a solid methodology, it is comprehensively written, well discussed and strength and limitations are clearly indicated. I have only minor comments to make.

Authors’ reply: We thank the reviewer for the words of appreciation.

Specific comments:

Reviewer’s comment: Line 49: Does the Nutrient Profiling Score identify products as “healthier” or “less healthy” or is only the category “less healthy” defined by the profiling model? Specify the wording.

Authors’ reply: We have changed the wording of the sentence to take into account the reviewer’s comment:

“The NPS considers the balance of fiber, fruit and vegetable, and protein content against energy, sugars, saturated fat, and salt (sodium) content. Based on their NPS, products can then be defined as "healthier" or "less healthy". The NPS is currently used to identify the products that cannot be advertised during children's television programs (i.e. if products have a NPS above a specified threshold, they are deemed ‘less healthy’ and are not permitted to be advertised to children).”

Reviewer’s comment: Line 118: Check font size

Authors’ reply: Thanks, this has been modified.

Reviewer’s comment: Line 263: Both.. and consumers – not clear. Separate the two aspects, the manufacturer can alter the product composition, the consumer can only alter the meal composition during cooking

Authors’ reply: Many thanks for the useful comment. As our analysis focuses on manufactured products, we deleted the word “consumers” and left only the word manufacturers.

Reviewer’s comment: Line 273 – 283: plain poultry will be salted during cooking/ at the table in contrast to plant-based plain poultry. This could explain the differences in salt content. However, would a consumer individually salt plain poultry less than the processed meat alternative?

Authors’ reply: This is a very interesting point. We agree that consumers add salt to plain poultry products during home-cooking but it is impossible to determine with precision how much salt they add to their dishes. Most of the available evidence on the topic refers to the salt content of manufactured products which provide approximately three quarters of the salt in UK diets. Plain poultry products such as chicken breast is a product typically consumed by health-conscious individuals and it is possible to assume that a home-cooked chicken breast would be less salty than a plant-based chicken product. Our results show that on average a plant-based chicken product sold in the UK would provide 1.3g salt per 100g. This amount of salt is equivalent to a quarter of the maximum salt daily recommendation (i.e. 5g/day).

Reviewer’s comment: Line 302 – limitations: it is very important that you mentioned the potential differences in micronutrient content. However, you should also add the aspect of protein nutritional quality e.g. protein digestibility-corrected amino acid score and digestible indispensable amino acid score.

Authors’ reply: We added to the limitation section the results of a recent study reporting that in the US, plant-based minced beef had lower levels of vit B12 and zinc. We also added that our results focus on those nutrients critical for NCD prevention.

“To evaluate overall product healthfulness, we used the NPM and the UK FoP labeling, which focus on crucial nutrients for NCD prevention and do not take into account micronutrient content. Meat is a source of important micronutrients such as zinc, iron, and vitamin B12. A recent investigation carried out in the US reported that plant-based minced beef has lower levels of zinc and vit B12”.

Reviewer’s comment: Line 341: last punctuation mark is missing

Authors’ reply: Thanks, this has been addressed.